# Erbium-Doped LiYF$_4$ as a Potential Solid-State Frequency Reference: Eligibility and Spectroscopic Assessment

Erik Cerrato [1], Chiara Gionco [1,*], Giuseppe Rizzelli Martella [2,3], Cecilia Clivati [1], Roberto Gaudino [3] and Davide Calonico [1]

1. Istituto Nazionale di Ricerca Metrologica (INRiM), Strada delle Cacce 91, 10135 Torino, Italy; e.cerrato@inrim.it (E.C.); c.clivati@inrim.it (C.C.); d.calonico@inrim.it (D.C.)
2. Fondazione LINKS, Via P. C. Boggio 61, 10138 Torino, Italy; giuseppe.rizzelli@polito.it
3. Dipartimento di Elettronica e Telecomunicazioni (DET), Politecnico di Torino, Corso Duca degli Abruzzi 24, 10129 Torino, Italy; roberto.gaudino@polito.it
* Correspondence: c.gionco@inrim.it

**Abstract:** Time and frequency metrology is a key enabler for both forefront science and innovation. At the moment, atomic frequency standards (AFSs) are based on atoms either in the vapor phase or trapped in magneto-optical lattices in a vacuum. Finding a solid-state material that contains atoms suitable to be used as a frequency reference would be an important step forward in the simplification of the setup of AFSs. Lanthanide-doped inorganic crystals, such as Er-doped LiYF$_4$, have been studied for several decades, and their intrashell 4f transitions are usually identified as ultra-narrow. Nevertheless, a systematic characterization of these transitions and their linewidths with a correlation to the dopant's concentration and isotopic purity at low temperatures is lacking. In this work, we studied Er-doped LiYF$_4$ as a potential benchmark material for solid-state frequency references. We chose Er as it has a set of transitions in the telecom band. The influence of Er concentrations and isotope purity on the transition linewidth was systematically studied using high-resolution optical spectroscopy at 5 K. The results indicate that the material under study is an interesting potential candidate as a solid-state frequency reference, having transition linewidths as low as 250 MHz at ~1530 nm.

**Keywords:** rare-earth ions; single crystal; atomic frequency reference

## 1. Introduction

Currently, state-of-the-art atomic frequency standards (AFSs) are the most accurate representation of a physical quantity. Atomic clocks based on the hyperfine transition of the cesium-133 ground state in the microwave domain, which is the current definition of the second according to the International System of Units, reach fractional frequency uncertainties in the order of $10^{-16}$. AFSs based on atomic transitions in the optical domain, so-called optical clocks, have surpassed cesium standards and presently offer fractional frequency uncertainties of parts at the $10^{-18}$ level [1–4]. However, such systems are very large and complex, requiring multiple lasers to cool the atoms or ions, and as a result, they are usually operated only at national metrology institutes or specialized laboratories. Research on transportable, compact, and miniaturized optical clocks has flourished in recent decades, reaching instabilities at 1 s of the order of $10^{-11}$ for miniaturized systems [5–7]. Analysis in several papers shows that the clock's performance and its volume are proportional [8].

Identifying a frequency reference in the solid state might break this correlation. Indeed, even at a low doping concentration, the number of atoms or ions in a solid matrix is several orders of magnitude higher than that of gaseous systems or optical lattices. Consequently, a solid-state AFS would have a better signal-to-noise ratio as this property scales with the number of entities, resulting in better stability. Moreover, the atomic species in high-performance AFSs need to be prepared using multiple single-frequency lasers, resulting in

significant dead time between subsequent interrogations. On the contrary, in a solid-state AFS, the atomic species are prepared during matrix synthesis, so there would be no dead time during operation, and a single interrogation laser would be sufficient. Finally, solid-state materials can be realized with processes compatible with chip fabrication, allowing for the optoelectronic integration of the device and leading to a higher miniaturization level. However, some challenges are associated with the realization of a solid-state AFS. The first one is linewidth broadening, both homogeneous and inhomogeneous, resulting in low-quality factor resonances. The second one is related to the presence of the matrix crystal field, which results in an additional perturbation of the atomic states and thus leads to frequency shifts in the atomic transitions that depend on both the quality and type of the solid-state matrix.

For this reason, the host matrix should be a single crystal with very low stress and strain. Among the available synthesis techniques used to obtain fluoride single crystals, the Czochralski method is a well-established one. With this technique, precursor powders are melted at high temperatures in a furnace under an argon atmosphere to prevent oxidation. Then, the crystallization is induced via dipping an oriented seed (typically an undoped crystal), and the growing crystal is withdrawn upwards at a controlled pulling and rotating speed [9–11].

Lanthanides are the 15 metallic elements that range from lanthanum to lutetium and are characterized by the (partial) filling of the 4f subshell. These electrons are shielded from the environment via 5s and 5p closed shells. Therefore, 4f intrashell transitions are basically insensitive to external perturbations, providing a good platform for an AFS.

The $4f^N$ energy level states are usually described under the Russell–Saunders scheme by J multiplets $^{2S+1}L_J$, where S, L, and J denote the electron spin, orbital, and total (vectorial sum of S and L) angular momenta in units of $\hbar$, respectively [12]. Even for free ions, some admixing of different S L states is possible, and it is determined via the spin–orbit coupling constant $\zeta_{4f}$, which increases along with the series.

When lanthanide ions ($Ln^{n+}$) are hosted in a crystalline matrix, the energy levels split further due to the effect of the crystal field, the number of levels depending on the value of J, and the hosting site's point group symmetry. In general, the number of levels increases with J when lowering the site symmetry. The mixing of J levels can change the levels' energy [12]. The crystal field contribution to the Hamiltonian, $\hat{H}_{CF}$, depends on the ion site's point symmetry via crystal field (CF) parameters.

The absorption and emission lines are homogeneously broadened via phonons at room temperature, with typical linewidths of tens to hundreds of GHz. On the other hand, at cryogenic temperature, the phonon contribution is lowered, and the homogenous linewidth ($\Gamma_h$) is limited by the excited state's lifetime, enabling $\Gamma_h$ values as low as 70–100 Hz below 4 K. At this temperature, the main contribution to broadening is the inhomogeneous one ($\Gamma_{inh}$). $\Gamma_{inh}$ is caused by the fact that each optical center experiences a slightly different host material environment. The cause of this can be some strain in the crystal, defects, impurities, and so on. Typical $\Gamma_{inh}$ values for REI-doped crystals span from 0.5 to 100 GHz, but values as low as 16 MHz have been reported for peculiar systems [13–19].

Homogeneous linewidths as low as 70–100 Hz have been reported when using a technique called spectral hole burning (SHB) [20–22]. This technique, although promising for laser stabilization, does not provide a significant simplification of the frequency reference since the spectral holes must be written in the material (with light pulses) and their lifetime is limited. High stability has been demonstrated but only for less than ten seconds; then, the hole deteriorates, and the crystal must be heated to be restored to the original state, with significant dead time.

Hence, a solid-state AFS should keep the simple setup of the SHB approach, tackling its lack of reliability and exhibiting similar ultra-narrow linewidths, or at least a fair trade-off between reliability and linewidths in order to achieve a significant instability both in the short and medium term.

This paper presents an assessment of erbium-doped lithium yttrium fluoride (Er:LiYF$_4$) single crystals as a candidate material for the implementation of a solid-state atomic frequency reference without the use of spectral holes. First, different combinations of dopant and matrix are discussed to justify the system's choice. Then, the system is systematically studied using high-resolution transmission optical spectroscopy to assess the influence of concentration and isotopic purity on the studied transitions.

## 2. Materials and Methods

### 2.1. Material Design

The ideal REI for building a frequency reference exhibits narrow light absorption in the NIR or visible range due to 4f-4f electronic transitions. Transitions in the C or O telecom regions (1550 and 1350 nm, respectively), similarly to those of Er (III) and Nd (III), are attractive since lasers, fibers, and other optical equipment that work at these frequencies are readily available and have a relatively high level of integration at the chip scale. The isotopic composition of the ensemble is also crucial. Indeed, the presence of several isotopes can contribute significantly to the linewidth's broadening due to the fluctuation of the electric field perceived by the electron cloud at different sites. Besides the number of isotopes, their nuclear spin should be considered. Indeed, a non-zero nuclear spin introduces a non-zero magnetic moment, increasing the inhomogeneity between sites. Finally, it is crucial to consider the number of f electrons: if it is odd (Kramers ions), each substate is at least double-degenerate, with the degeneracy removed via the presence of a magnetic field [23]; thus, it has increased sensitivity relative to the environment. When assessing the potential host matrix crystals, the desirable features are as follows:

- The radius of the cation to be substituted should be as close as possible to the REI radius (for the same coordination) in order to minimize strains due to doping, especially when considering higher concentrations;
- The cation to be substituted and the REI should have the same oxidation state to avoid charge compensation;
- The cation to be substituted should occupy a single crystalline site (i.e., all sites should be equivalent);
- A cationic site with high symmetry to minimize the crystal field's splitting;
- Minimum magnetic moment; hence, elements with null or low magnetic spin are preferable;
- Isotropic linear coefficient of thermal expansion (CTE) to avoid excessive deformation during cooling;
- The lowest possible presence of defects; hence, its high purity that is both chemical and isotopic, and the best crystalline quality are obtained;
- Synthesis methods compatible with standard device fabrication techniques must be considered a plus.

Given these considerations, we considered several possible host matrices: Y$_2$O$_3$, Y$_2$SiO$_2$, YVO$_4$, Y$_3$Al$_5$O$_{12}$, CaF$_2$, SrF$_2$, BaF$_2$, LiYF$_4$, SrTiO$_3$, TiO$_2$, ZrO$_2$, HfO$_2$, ZnO, Al$_2$O$_3$, SiO$_2$, CaWO$_4$, SrWO$_4$, BaWO$_4$, and Y$_2$Ti$_2$O$_7$.

Since the lowest possible presence of defects is of paramount importance, a cation with a similar radius to the REI and the same oxidation state is the first discriminant for the matrix's choice. Transition metal ions usually have ionic radii that are significantly smaller than the lanthanide ones (60–80 pm vs. 98–114 pm), and they often have oxidation states different from +3 or multiple oxidation states that are easily interchangeable (high redox activity). Alkaline earth atoms have large radii that can accommodate the REIs, but a +2 oxidation state indicates that charge compensation defects would arise. Aluminum has the correct charge, but it has a small radius that would induce high crystal strain.

Therefore, we deduced that materials containing Y appear as the most appropriate for the preparation of REI-doped crystals. Indeed, Y has the same charge and a similar radius to several REIs, especially in the second half of the series [24]. It has a single stable isotope with a small, although not null, magnetic moment [25]. A drawback of Y is that

it is vicariant with several rare earths; thus, it is impossible to obtain perfectly undoped materials and control a very low percentage of doping. Nevertheless, Y-containing materials are the most popular matrices for REI-doped crystals relative to various applications, with LiYF$_4$ (LYF) [13,17,26,27], Y$_2$SiO$_5$ (YSO) [14,20,28,29], and Y$_2$O$_3$ [25,30,31] being the most used ones. Among these, LYF is the only one where Y occupies a single crystalline site. Although F has a magnetic moment, a single isotope is present with a small nuclear spin (1/2). Moreover, REI-doped LYF is readily available from laser suppliers as a single crystal with very low defectivity, and a decent amount (although non-exhaustive) of literature already exists to compare data as a starting point.

The LYF crystalline structure belongs to space group I4$_1$/a; hence, it has a body-centered tetragonal unit cell with a screw axis and glide plane. Y, and therefore the substitutional lanthanide, occupies a single site with S4 symmetry [10].

Therefore, we address Er-doped LYF as a benchmark material, and in this study, we provide experimental evidence for its possible application as a solid-state frequency reference.

The concentration of Er ions should be chosen carefully. Indeed, a lower concentration would minimize the transition linewidth as the interaction between different ions is suppressed as the number of different sites. On the other hand, the concentration should be sufficient to obtain a good signal-to-noise ratio, as this should scale proportionally to the root square of the interrogated atoms. Considering that the interrogated atoms are in the order of $10^{15}$–$10^{16}$ even for very low concentrations, we chose three samples:

- S1: $^{166}$Er:$^7$LiYF$_4$ with Er concentration of 100 ppm (molar fraction);
- S2: Er:LiYF$_4$ with Er concentration of 100 ppm (molar fraction);
- S3: Er:LiYF$_4$ with Er concentration of 35 ppm (molar fraction).

In this way, we can assess the effect of both isotopic purity and REI concentrations on the resonance frequency and linewidth of the optical transitions of interest.

### 2.2. Material Realization and Experimental Setup

Single crystals $^{166}$Er:$^7$LiYF$_4$ with Er concentrations of 100 ppm (S1) and Er:LiYF$_4$ (natural abundance) with Er concentrations of 100 ppm (S2) and 35 ppm (S3) were purchased from MEGA Materials (Pisa, Italy) and employed without any additional annealing treatment. The samples are 4 × 4 × 4 mm$^3$.

Samples were grown using the Czochralski (CZ) technique. Errors on the precursors' weighting are in the order of 5%; therefore, the maximum error relative to Er concentrations is 10%. Monocristallinity was checked using X-ray diffraction, which was also used for crystal orientation before cutting the samples. Samples were finally polished to spectroscopic grade quality on four facets out of six.

High-resolution transmission spectra were recorded via the setup shown in Figure 1 and mounted on an anti-dumping optical table. The near-IR light source was an 81980A Compact Tunable Laser Source (Keysight Technologies, Santa Rosa, CA, USA) with a linewidth of 100 kHz; the scans were performed with a wavelength resolution of 1 pm and a time per step of 5 s. The irradiation power at the sample surface was 500 μW and had a light spot of 1.3 mm. The samples were mounted inside a Montana Instrument closed-cycle optical cryostat (S-100, temperature stability <15 mK), with the a-axis perpendicular to the laser beam, and thermal grease was used to guarantee thermal contact. The transmitted light was monitored via an InGaAs photodiode operating in the air and connected to a mixed signal oscilloscope. Light polarization was set as transverse electric (TE). The entire measurement was automated using MATLAB R2021b software, Natick, MA, USA, while data analysis was performed employing Origin 2022 software, Boston, MA, USA.

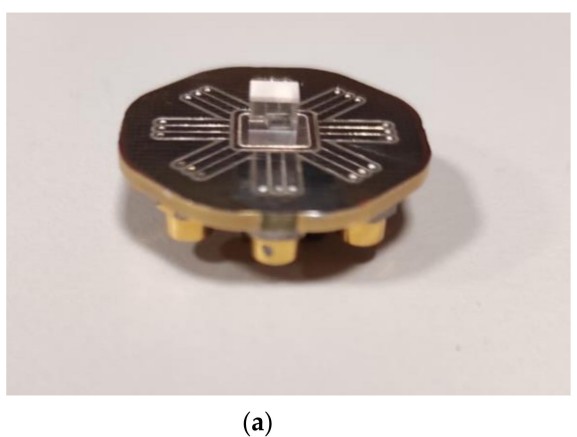
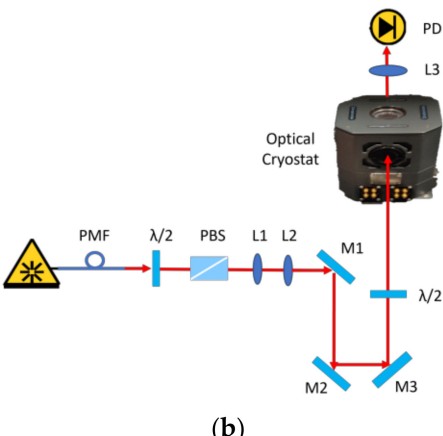

(**a**)                                                          (**b**)

**Figure 1.** (**a**) A picture of sample $^{166}$Er:$^7$LiYF$_4$ 100 ppm. (**b**) Measurement setup. PMF: polarization-maintaining fiber; L: lens; M: mirror; $\lambda/2$: polarization rotator; PBS: polarization beam splitter; PD: photodiode. The blue lines represent the optical fiber path, and the red ones represent the free space path.

### 3. Results

*High-Resolution Spectroscopy*

To assess whether our samples can be considered a benchmark material for a solid-state frequency reference, we studied ultra-narrow optical transitions from the lowest ground state ($^4$I$_{15/2}$) to the first excited state ($^4$I$_{13/2}$). When the lanthanide ions (Ln$^{n+}$) are hosted into a crystalline matrix, energy levels split further due to the effect of the crystal field, with the number of levels depending both on the value of J and the hosting site's point group symmetry. In general, the number of levels increases with J, lowering the site symmetry. The admixing of J levels can change the levels' energy [12]. The crystal field contribution to the Hamiltonian, $\hat{H}_{CF}$, depends on the ion site's point symmetry relative to crystal field (CF) parameters. An electronic transition is potentially allowed or forbidden according to specific selection rules. It is possible to apply the SLJ selection rules from the Judd–Ofelt theory [32,33] to 4f–4f transitions. Intrashell f–f transitions are electric dipole (ED)-forbidden (there is no change in parity) but can be forced via an induced electric dipole (Judd forced) for specific values of the SLJ vectors, with strength at 2–3 orders of magnitude lower than the ones with a change in parity. In addition, magnetic dipole (MD) transitions can be allowed with a strength lower than two orders of magnitude. Since the intrashell transitions are MD or second-order ED, their oscillator strength is relatively weak, enabling long excited state lifetimes (up to 10 ms). Nevertheless, thanks to their narrow nature, good optical densities can be achieved [13].

When the ions are ligand-coordinated, the site's point group selection rules also apply. These selection rules consider the irreducible representations of the initial and final crystal field states ($\Gamma_i$ and $\Gamma_f$) and that of the electric and magnetic dipole moment operators ($\Gamma_{ED}$ and $\Gamma_{MD}$). As a result, some transitions may be allowed by SLJ rules but forbidden by the site's point group [12].

In the LYF crystal structure, the Er$^{3+}$ ion occupies a site with S4 point symmetry; as a consequence, the ground state splits into eight Stark states while the first excited state splits into seven Stark states, as depicted in Figure 2, where each sublevel is identified with the irreducible representation based on the selection rules for electric-dipole (ED) and magnetic-dipole (MD) transitions between Stark levels in tetrahedral symmetry. Since Er$^{3+}$ is a Kramers ion (having an odd number of electrons), each sublevel has double degeneracy.

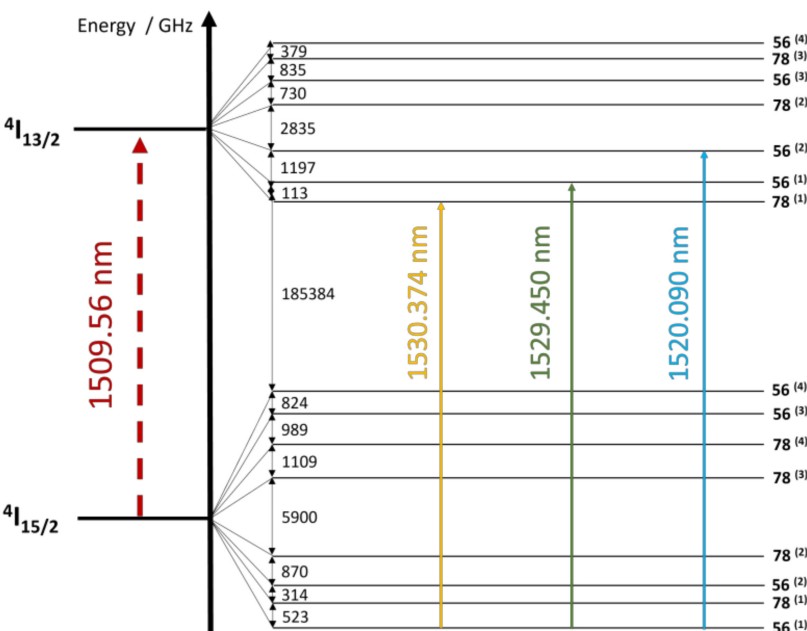

**Figure 2.** Ground state ($^4I_{15/2}$) and first excited state ($^4I_{13/2}$) $4f^{11}$ energy level diagram for $Er^{3+}$:LiYF$_4$. Yellow, green, and blue solid arrows indicate the investigated optical transitions T1, T2, and T3, respectively. Splitting are given in GHz. The notation on the right identifies the irreducible representation based on the selection rules for electric dipole (ED) and magnetic dipole (MD) transitions between Stark levels in tetrahedral symmetry.

In this work, we studied the optical transitions from the lowest ground state ($^4I_{15/2}$) doublet $\Gamma'_{56}{}^{(1)}$ to the lower three manifolds of the first excited state: ($^4I_{13/2}$) $\Gamma'_{78}{}^{(1)}$ (from now on, T1), $\Gamma'_{56}{}^{(1)}$ (T2), and $\Gamma'_{56}{}^{(2)}$ (T3); they are indicated by arrows in Figure 2, and they are centered at 1530.374 nm (195.89490 THz), 1529.450 nm (196.01325 THz), and 1520.09 nm (197.22059 THz). Indeed, recording the spectra at 5 K, we can assume that only the lowest ground state ($^4I_{15/2}$) doublet $\Gamma'_{56}{}^{(1)}$ is populated; thus, we consider it as the only initial state [34–36].

The absorption spectra recorded at 5 K for S1 are reported in Figure 3.

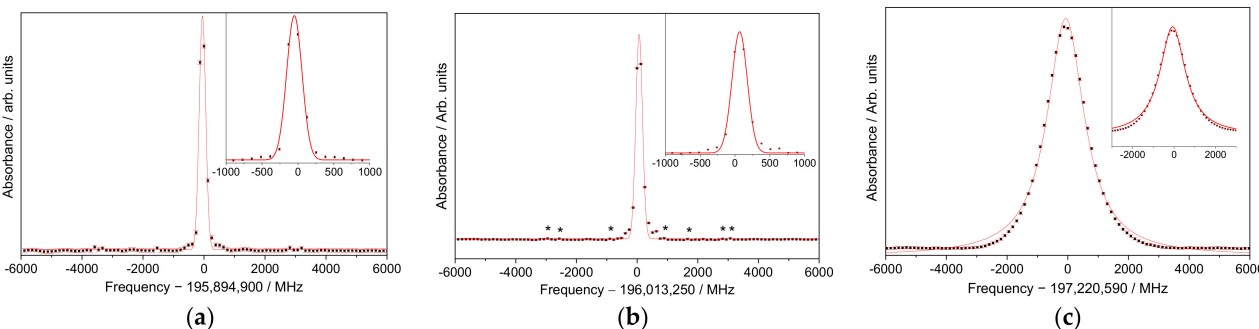

**Figure 3.** $Er^{3+}$ absorption spectra and their fitting of sample $^{166}Er$:$^7$LiYF$_4$ 100 ppm for (**a**) $^4I_{15/2}$ $\Gamma'_{56}{}^{(1)}$ $\rightarrow$ $^4I_{13/2}$ $\Gamma'_{78}{}^{(1)}$; (**b**) $^4I_{15/2}$ $\Gamma'_{56}{}^{(1)}$ $\rightarrow$ $^4I_{13/2}$ $\Gamma'_{56}{}^{(1)}$; (**c**) $^4I_{15/2}$ $\Gamma'_{56}{}^{(1)}$ $\rightarrow$ $^4I_{13/2}$ $\Gamma'_{78}{}^{(2)}$. Asterisks indicate absorption due to the $^{167}Er$ isotope.

Considering S1, the spectra each appear as a narrow single line centered at 1530.37(4) nm for the first transition, 1529.44(9) nm for the second transition, and 1520.08(7) nm for the third transition. In addition to the main line, further tiny absorptions are also observable (asterisks in the figure) on both sides, arising from the unavoidable presence of the $^{167}Er$ isotope. Indeed, although the sample is isotopically pure in $^{166}Er$, the vicariance of this element with Y accidentally introduces an uncontrolled amount of Er isotopes, among

which $^{167}$Er spectral components can be easily recognized due to its peculiar spectral features coming from ground-state (and excited) Kramers doublets, which are split in a complex pattern via the hyperfine interaction [37]. These spectroscopic features do not represent a limitation to the intended application since they are of negligible intensity with respect to the main peak. For this first sample, fitting a single Gaussian peak with a full-width-at-half-maximum (FWHM) value of 252 ± 4 MHz for the first transition, 258 ± 3 MHz for the second transition, and 1486 ± 23 MHz for the third transition is sufficient, and the fit parameters are reported in Table 1.

**Table 1.** Parameters obtained from the Gaussian fit for the three investigated samples.

| Transition | Sample | Peak Centers [1,2,3]/MHz | | | FWHM */MHz | R[2,†] |
| --- | --- | --- | --- | --- | --- | --- |
| | | Peak 1 | Peak 2 | Peak 3 | | |
| T1 [1] | S1 | −50.3 ± 1.5 | / | / | 252 ± 4 | 0.99 |
| | S2 | −71 ± 4 | 467 ± 8 | 1050 ± 40 | 397 ± 7 | 0.99 |
| | S3 | −80 ± 2 | 427 ± 9 | 760 ± 60 | 333 ± 3 | 0.99 |
| T2 [2] | S1 | 66 ± 1 | / | / | 258 ± 3 | 0.99 |
| | S2 | 78 ± 9 | 653 ± 15 | 1200 ± 20 | 480 ± 13 | 0.99 |
| | S3 | 19 ± 3 | 487 ± 6 | 927 ± 25 | 330 ± 5 | 0.99 |
| T3 [3] | S1 | −68 ± 6 | / | / | 1486 ± 23 | 0.99 |
| | S2 | 128 ± 16 | / | / | 3490 ± 70 | 0.99 |
| | S3 | −37 ± 10 | / | / | 1571 ± 34 | 0.99 |

* FWHM was shared among the contributions where more than one peak was present; [1] 195,894,900 MHz was added to peak center values; [2] 196,013,250 MHz was added to peak center values; [3] 197,220,590 MHz was added to peak center values; [†] coefficient of determination, determined as $R^2 = 1 − $ (residual sum square/total sum square).

The absorption spectra recorded at 5 K for S2 and S3 are reported in Figures 4 and 5, respectively.

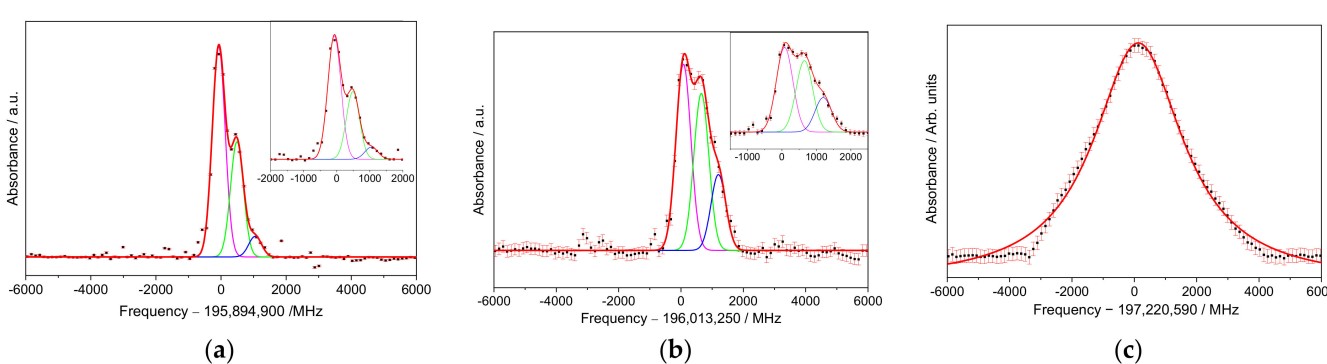

(a)        (b)        (c)

**Figure 4.** Er$^{3+}$ absorption spectra and their fitting of sample Er:LiYF$_4$ 100 ppm for (**a**) $^4$I$_{15/2}$ $\Gamma'_{56}$$^{(1)}$ → $^4$I$_{13/2}$ $\Gamma'_{78}$$^{(1)}$; (**b**) $^4$I$_{15/2}$ $\Gamma'_{56}$$^{(1)}$ → $^4$I$_{13/2}$ $\Gamma'_{56}$$^{(1)}$; (**c**) $^4$I$_{15/2}$ $\Gamma'_{56}$$^{(1)}$ → $^4$I$_{13/2}$ $\Gamma'_{78}$$^{(2)}$. The red line is the total nonlinear fitting, the magenta, green and blue lines are the individual peaks that contribute to the convolution.

When considering samples S2 and S3, where all natural Er and Li isotopes are present, the spectra are more complex, and a multi-peak fitting is necessary. For transitions T1 and T2 and both samples, the best fit was obtained when using three Gaussian peaks and sharing their FWHM, as evidenced in Table 1. The three peaks can probably be associated with the presence of the even isotopes of Er [16,17,38]; nevertheless, the ratio between the areas of the resulting peaks is not always comparable to the isotopes' natural abundances. Thus, further studies are required for a conclusive assignment. For transition T3, the line was broadened at the point at which different contributions were not resolvable, and a single Gaussian peak was used to fit the spectra.

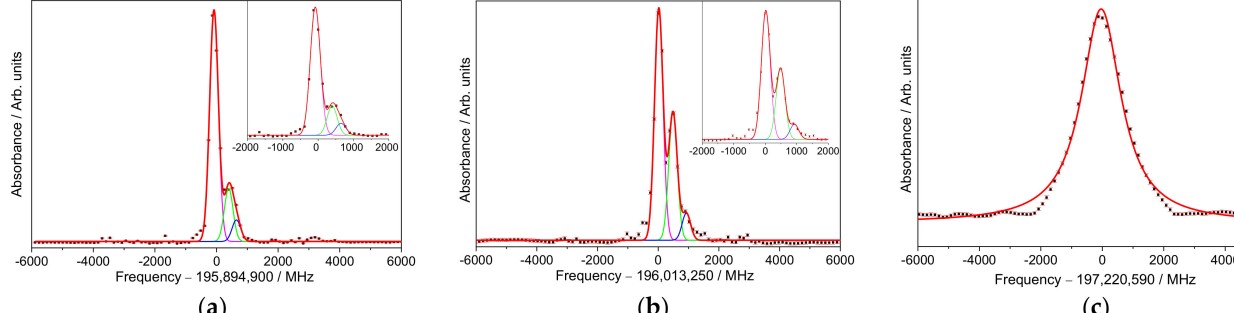

**Figure 5.** $Er^{3+}$ absorption spectra and their fitting of sample Er:LiYF$_4$ 35 ppm for (**a**) $^4I_{15/2}$ $\Gamma'_{56}{}^{(1)}$ $\rightarrow$ $^4I_{13/2}$ $\Gamma'_{78}{}^{(1)}$; (**b**) $^4I_{15/2}$ $\Gamma'_{56}{}^{(1)}$ $\rightarrow$ $^4I_{13/2}$ $\Gamma'_{56}{}^{(1)}$; (**c**) $^4I_{15/2}$ $\Gamma'_{56}{}^{(1)}$ $\rightarrow$ $^4I_{13/2}$ $\Gamma'_{78}{}^{(2)}$. The red line is the total nonlinear fitting, the magenta, green, and blue lines are the individual peaks that contribute to the convolution.

Figure 6 reports the effect of temperature (5 K, 10 K, and 20 K) on T1 for the three samples. To allow a direct comparison, the absorbance spectra were normalized to 1, and the fitting was added to guide the eye.

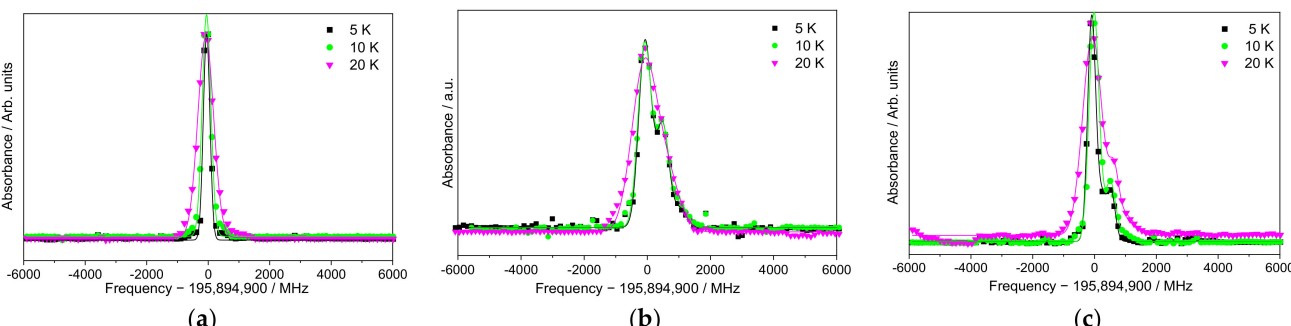

**Figure 6.** $Er^{3+}$ absorption spectra for transition $^4I_{15/2}$ $\Gamma'_{56}{}^{(1)}$ $\rightarrow$ $^4I_{13/2}$ $\Gamma'_{78}{}^{(1)}$ at 5, 10, and 20 K for samples (**a**) $^{166}Er$:$^7LiYF_4$ 100 ppm; (**b**) Er:LiYF$_4$ 100 ppm; (**c**) Er:LiYF$_4$ 35 ppm.

The main effect of increasing the temperature is, as expected, the broadening of the peak. Nevertheless, it is worth noting that the difference between 5 and 10 K is minimal.

## 4. Discussion

The reported results provide systematic insight into the effects of isotopic purity, dopant concentration, and temperatures, confirming some of the assumptions made in Section 2.1.

For instance, the effect of Er ion concentrations on the transition linewidth can be extrapolated from Table 1. For both T1 and T2, the linewidth of S2 is 1.2–1.4 times that of S3. A linear fit of the combined points from T1 and T2 for S2 and S3 suggests an effect of $(1.6 \pm 0.6)$ MHz/ppm with a contribution of $(274 \pm 48)$ MHz independent of the ions' concentration. Since concentration-related broadening can be explained as the perturbation induced via the interaction between dopant ions, we can interpret this as the inhomogeneous linewidth's broadening, which is related exclusively to the matrix and the crystal field.

Similarly, at a fixed concentration, the use of single isotopes for both Er and Li provides a narrowing of the absorption peak by 1.5–1.9 times, confirming the key role of the presence of several isotopes in the linewidth's broadening. Indeed, the presence of different isotopes creates a difference in the local field experienced by the dopant ions and, therefore, slightly shifts their energy levels with respect to each other.

Unfortunately, the obtained SNR is significantly lower than the expected value and is calculated relative to the number of interrogated atoms ($10^2$ instead of $10^8$), indicating that

limits arising from the experimental setup are present and should be addressed to unlock the full potential of these systems. The source of this limit and the relative overcome will be studied in the near future.

The obtained resonance quality factor is in the order of 4 to $8 \times 10^5$, which is relatively high for a solid-state device. Although devices with higher Q factors exist in the solid state [39–41], these are not directly related to atomic transitions, similarly to the one described in the present paper. By engineering the Er concentration and using isotopic pure samples, Q factors of at least one order of magnitude higher are likely achievable. When combining a higher Q factor with the optimization of the setup to maximize the signal-to-noise ratio, it can be envisaged that a solid-state platform for the laser stabilization of atoms using rare-earth-doped inorganic crystals can be realized.

Finally, it is worth noting that the obtained linewidths are very similar for both 5 and 10 K, slightly relaxing the low-temperature requirement of the system. Research providing a deeper insight into temperature dependence is currently ongoing.

## 5. Conclusions

We studied $Er:LiYF_4$ as a potential material for ultra-stable solid-state atomic frequency standards. Since the inhomogeneous transition linewidth is an important feature for this kind of application, we spectroscopically investigated three different samples of single-crystal $Er^{3+}$-doped $LiYF_4$ at cryogenic temperatures. Three distinct transitions were assessed, corresponding to the excitation from the ground level ($^4I_{15/2}$)'s lowest Stark state and the three lowest Stark levels of the first excited state of $Er^{3+}$ ($^4I_{13/2}$). The low dopant concentration and the similarity in the dopant and substituted ion radii allowed the growth of crystals with low strain, resulting in inhomogeneous linewidths recorded as low as ~250–300 MHz at ~1530 nm for both the sample with the lowest Er concentration (35 ppm) and the isotopically pure sample with higher concentrations (100 ppm). We found an effect of Er concentration on the linewidth of $(1.6 \pm 0.6)$ MHz/ppm, with a contribution of $(274 \pm 48)$ MHz independent of the ions' concentration. The transitions of the samples containing not isotopically purified Er and Li were fitted using three peaks, indicating a probable contribution from the different even isotopes of Er (166, 168, and 170), although further studies are necessary for a conclusive assignment since the peaks area do not match with the natural abundance of isotopes. The temperature dependence of the center frequency and linewidth is currently under investigation. This also holds true for the setup's optimization in order to increase the signal-to-noise ratio to its maximum. Although the present results do not reach values that could surpass current miniaturized frequency references, they are still encouraging; therefore, further studies including effective laser stabilization relative to the atoms will be performed in the future after the signal-to-noise ratio's optimization.

**Author Contributions:** Conceptualization, C.G. and D.C.; methodology, E.C., C.G. and C.C.; software, G.R.M.; investigation, E.C.; data curation, E.C. and C.G.; writing—original draft preparation, C.G. and E.C.; writing—review and editing, all authors; funding acquisition, D.C. and R.G. All authors have read and agreed to the published version of the manuscript.

**Funding:** This research was funded by the Piemonte Region within the "Infra-P" scheme (POR-FESR 2014–2020 program of the European Union) project "Piemonte Quantum Enabling Technologies" (PiQuET) and by the Fondazione Cassa di Risparmio di Torino (grant number: 104370/2023.0400).

**Data Availability Statement:** The data presented in this study are available upon request from the corresponding author.

**Conflicts of Interest:** The authors declare no conflict of interest. The funders had no role in the design of this study; in the collection, analyses, or interpretation of data; in the writing of the manuscript; or in the decision to publish the results.

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
