# Peer review of "Erbium-Doped LiYF4 as a Potential Solid-State Frequency Reference: Eligibility and Spectroscopic Assessment"

_crystals, doi:10.3390/cryst13101476_

Round 1

Reviewer 1 Report

This work is devoted to the study of doped fluoride LiYF4. Systems of the A(I)B(III)Hal4 type and doped compositions based on them have been studied for several decades. A huge amount of data on this group of compounds has been accumulated and discussed in the open press. However, studies on the fine structure of the spectra of rare earth ions in well-studied matrix media are welcome. From this point of view, I consider the study relevant.

While the article is generally well written, there are a number of issues that need to be addressed before publication.

1. The authors do not provide information about methods for the synthesis of alloyed compounds. There is no information about their phase composition, degree of purity, or dispersity.

2. The section materials and methods needs to be substantially reduced in the part that is devoted to the description of equipment for spectral measurements. Figure 1 should generally be removed or moved to supporting materials.

3. Sections 3 and 4 should be combined, giving consecutive discussions for each of the most significant results. In this form, the article does not look complete.

4. The abstract of the article must be brought to general standards. It sets out only the results achieved in a "dry" form. Without introductory discussions about the current state of science and its prospects. These arguments are more suitable for a conclusion. But all the measured parameters from the conclusion should just be transferred to the annotation. It seems that the annotation and conclusion need to be swapped.

5. Unfortunately, I did not find crystal-chemical and crystal-physical inference in the Conclusion. In this form, the article is more suitable for a specialized journal on optics. Either the authors should fix it.

Unfortunately, I am forced to submit the article for major revision or recommend it for a more specialized optical journal.

Author Response

This work is devoted to the study of doped fluoride LiYF4. Systems of the A(I)B(III)Hal4 type and doped compositions based on them have been studied for several decades. A huge amount of data on this group of compounds has been accumulated and discussed in the open press. However, studies on the fine structure of the spectra of rare earth ions in well-studied matrix media are welcome. From this point of view, I consider the study relevant.

We thank the reviewer for the constructive comments, we updated the manuscript accordingly whenever possible.

While the article is generally well written, there are a number of issues that need to be addressed before publication.

  1. The authors do not provide information about methods for the synthesis of alloyed compounds. There is no information about their phase composition, degree of purity, or dispersity.

Thanks for your comment. Indeed, as stated in the Materials section, the samples were not synthesized by us, but were purchased from a specialized company named “MegaMaterials s.r.l.”. Therefore, we have limited information on the preparation of the materials themselves, but we added all the information that we have in the manuscript.

  1. The section materials and methods needs to be substantially reduced in the part that is devoted to the description of equipment for spectral measurements. Figure 1 should generally be removed or moved to supporting materials.

We believe that the accurate description of the measurement setup is important in order to guarantee reproducible results, therefore we’d like to keep the description of the spectral measurement setup, nevertheless, we reduced the scheme in figure 1 and added a photo of the sample, and removed from the description the commercial names of the instruments used when they were standard ones.

  1. Sections 3 and 4 should be combined, giving consecutive discussions for each of the most significant results. In this form, the article does not look complete.

In this case, we followed the journal template which contemplate two separate sections for the results enunciation and their discussion. We asked the editor whether we should merge it as suggested but we were asked to stick to the journal authors’ guidelines (https://www.mdpi.com/journal/crystals/instructions#front).

  1. The abstract of the article must be brought to general standards. It sets out only the results achieved in a "dry" form. Without introductory discussions about the current state of science and its prospects. These arguments are more suitable for a conclusion. But all the measured parameters from the conclusion should just be transferred to the annotation. It seems that the annotation and conclusion need to be swapped.

We added to the abstract a sentence that clarify the current state of science and the work goal. For the abstract, we followed the instructions for authors of the journal, with a short background, a brief description of the methods used, the results and relative conclusions presented in an objective way, within the 200 words count.

  1. Unfortunately, I did not find crystal-chemical and crystal-physical inference in the Conclusion. In this form, the article is more suitable for a specialized journal on optics. Either the authors should fix it.

We modified the Conclusions to add some more information that was indeed available in the discussion section (the linewidth dependance on the dopant concentration), and tried to be more effective.

It is indeed a paper focused on the spectroscopic characterization of a crystalline material that justify a novel application for a well known material. From what we could infer from the “Aims” section of the journal, Emerging applications of inorganic crystalline materials as well as their optical properties are of interest for it. We leave of course the final decision to the editor.

Reviewer 2 Report

The paper presents the results of a study of ultra-narrow intra-4f junctions of Er-doped LiYF4 as a potential reference material for solid-state frequency references. In general, this area of research is very promising and interesting not only from a practical point of view, but more from a fundamental one, since this work is aimed at developing the applicability of new types of materials as standards. The results obtained have a certain level of scientific novelty and practical significance, which in the future can be used not only by the authors but also by the global scientific community. This article has very strong claims to the innovativeness of the proposed approach. In general, the work corresponds to the subject of the application and the journal and can be accepted for publication after the authors answer a number of questions from the reviewer.

1. In the introduction, the authors should pay attention and provide more data about the reasons for choosing LiYF4 as reference materials for creating solid-state reference frequencies, since the main goal of the article is precisely to study the prospects for using these materials as one of the subsequent standards.

2. The authors should provide a short comparative analysis of the changes they observed with other types of materials.

3. According to the presented spectral absorption lines (see data in Figure 3), a broadening of the spectral bands and a decrease in their intensity are clearly visible; the nature of such phenomena should be described in more detail, since broadening can play a very important role in this process.

4. The presented transition accuracy is very high, and therefore the authors should explain how exactly these values were obtained with this level of accuracy (up to the third decimal place).

5. In the abstract, the authors should mention the reasons for choosing Er as the alloying element, as well as what specific concentration dependences were considered when adding it to the LiYF4 composition.

Author Response

The paper presents the results of a study of ultra-narrow intra-4f junctions of Er-doped LiYF4 as a potential reference material for solid-state frequency references. In general, this area of research is very promising and interesting not only from a practical point of view, but more from a fundamental one, since this work is aimed at developing the applicability of new types of materials as standards. The results obtained have a certain level of scientific novelty and practical significance, which in the future can be used not only by the authors but also by the global scientific community. This article has very strong claims to the innovativeness of the proposed approach. In general, the work corresponds to the subject of the application and the journal and can be accepted for publication after the authors answer a number of questions from the reviewer.

We thank the reviewer for the constructive comments, we updated the manuscript accordingly whenever possible.

  1. In the introduction, the authors should pay attention and provide more data about the reasons for choosing LiYF4 as reference materials for creating solid-state reference frequencies, since the main goal of the article is precisely to study the prospects for using these materials as one of the subsequent standards.

The reasons for choosing LiYF4 as a matrix for the reference atoms (Er3+) are fully explained in section 2.1, Material Design. We added a couple of paragraphs to explain it further.

  1. The authors should provide a short comparative analysis of the changes they observed with other types of materials.

I’m not sure we fully understand the question, as this is the first material that we consider for the proposed application. State of the art frequency references use completely different setups with warm or cold vapors of atoms. In that case the atoms are closer to the unperturbed state and the achievable fractional uncertainties are very high, but the setups are very complicated.

  1. According to the presented spectral absorption lines (see data in Figure 3), a broadening of the spectral bands and a decrease in their intensity are clearly visible; the nature of such phenomena should be described in more detail, since broadening can play a very important role in this process.

Figure 3 presents three different transitions. The broadening observed is already known in literature and expected as the higher Stark levels are more easily coupled with the crystal phonons. For this reason the third transition is presented but not discussed further and the subsequent transitions are not investigated at all.

  1. The presented transition accuracy is very high, and therefore the authors should explain how exactly these values were obtained with this level of accuracy (up to the third decimal place).

As indicated in the materials and methods section, we used a 81980A Compact Tunable Laser Source (Keysight Technologies) as the near-IR light source with a linewidth of 100 kHz; the scans were performed with a wavelength resolution of 1 pm. We chose 1pm as it is the accuracy declared by the instrument manufacturer.

  1. In the abstract, the authors should mention the reasons for choosing Er as the alloying element, as well as what specific concentration dependences were considered when adding it to the LiYF4 composition.

We added to the abstract the main reason for choosing Er as the dopant element, but with this we reached the word count limit for the abstract (200 words).

Reviewer 3 Report

The paper entitled “Erbium-doped LiYF4 as a potential solid-state frequency reference: eligibility and spectroscopic assessment” contains a well-done work about the first steps for the preparation of a solid phase frequency reference. The paper is logically built, and the selection of LiYF4 material and erbium as dopant was done on a solid scientific basis. In general, the experiments and their explanations are adequate.

Due to the differences between the isotopically pure and natural isotope abundance samples (which was explained correctly), it would be useful to give the isotope composition of these samples (purity of the isotopically pure and isotope abundance for the natural isotope abundance sample) .

Author Response

The paper entitled “Erbium-doped LiYF4 as a potential solid-state frequency reference: eligibility and spectroscopic assessment” contains a well-done work about the first steps for the preparation of a solid phase frequency reference. The paper is logically built, and the selection of LiYF4 material and erbium as dopant was done on a solid scientific basis. In general, the experiments and their explanations are adequate.

Due to the differences between the isotopically pure and natural isotope abundance samples (which was explained correctly), it would be useful to give the isotope composition of these samples (purity of the isotopically pure and isotope abundance for the natural isotope abundance sample) .

We thank the reviewer for the constructive comments, we updated the manuscript accordingly whenever possible.

Unfortunately, we don’t have information to the analytical isotopic composition of the samples. Indeed, as stated in the Materials section, the samples were not synthesized by us, but were purchased from a specialized company named “MegaMaterials s.r.l.”. We asked them additional information, they used 7LiF from Sigma Aldrich as lithium and fluoride precursor, with 99% enrichment and 99% purity. They were not able to provide further information on the 166Er precursor. As far as it goes for the natural abundance materials, they usually use 5N precursor materials as well as 5N Argon in the furnace (see references 9-11).

Reviewer 4 Report

The article Erbium doped LiYF4 as a potential solid-state frequency reference: eligibility and spectroscopic assessment» makes a good impression and shows the possibility of using Er:LiYF4 single crystals in various devices, including modern atomic frequency standards (AFS). The article is written in good language and contains sufficient experimental data. However, the authors are recommended to make some adjustments to the presented material:

Part 2.1. in section “2. Materials and methods" it makes sense to move to section "1. Introduction”, because it contains general information and is not relevant to the experiment.

It would be nice to present a photo of the original crystal, and also briefly talk about the technology for producing Er:LiYF4 single crystals

Er:LiYF4 single crystals with different isotopic compositions and concentrations of Er3+ ions were used for the experiments. Was the actual erbium content in the resulting crystals controlled?

Author Response

The article “Erbium doped LiYF4 as a potential solid-state frequency reference: eligibility and spectroscopic assessment» makes a good impression and shows the possibility of using Er:LiYF4 single crystals in various devices, including modern atomic frequency standards (AFS). The article is written in good language and contains sufficient experimental data. However, the authors are recommended to make some adjustments to the presented material:

We thank the reviewer for the constructive comments, we updated the manuscript accordingly whenever possible.

Part 2.1. in section “2. Materials and methods" it makes sense to move to section "1. Introduction”, because it contains general information and is not relevant to the experiment.

The general information contained in section 2.1 has been moved to the Introduction, while the part relative to the scientific basis that justify the choice of the materials specific for the intended application has been retained in section 2.1 as it is part of the material design.

It would be nice to present a photo of the original crystal, and also briefly talk about the technology for producing Er:LiYF4 single crystals

The sample photo has been added to figure 1. As stated in the Materials section, the samples were not synthesized by us, but were purchased from a specialized company named “MegaMaterials s.r.l.”, therefore we don’t have photos of the boule but only of the cut samples.

We added a short paragraph in the introduction that describes the technology used for the samples production (although we are not specialized on this).

“For this reason, the host matrix should be a single crystal with very low stress and strain. Among the available synthesis techniques to obtain fluoride single crystals, the Czochralski method is a well established one. With this technique, the precursors powders are melted at high temperature in a furnace under Argon atmosphere to prevent oxidation. Then, then the crystallization is inducted by dipping an oriented seed (typically an undoped crystal) and the growing crystal is withdrew upwords at a controlled pulling and rotating speed”

Er:LiYF4 single crystals with different isotopic compositions and concentrations of Er3+ ions were used for the experiments. Was the actual erbium content in the resulting crystals controlled?

No, unfortunately the actual Er content was not controlled in the resulting crystals, nor by the supplier or by us. We don’t have any non-destructive (we are still working on the samples) analytical technique available to check such low concentrations, especially in the short timeline provided for the revision.

Round 2

Reviewer 1 Report

I recommend accepting the article for publication